# A new preprocedural predictive risk model for post-endoscopic retrograde cholangiopancreatography pancreatitis: The SuPER model

**Mitsuru Sugimoto[1]\*, Tadayuki Takagi[1], Tomohiro Suzuki[2], Hiroshi Shimizu[2], Goro Shibukawa[3], Yuki Nakajima[3], Yutaro Takeda[4], Yuki Noguchi[4], Reiko Kobayashi[4], Hidemichi Imamura[4], Hiroyuki Asama[5], Naoki Konno[5], Yuichi Waragai[6], Hidenobu Akatsuka[7], Rei Suzuki[1], Takuto Hikichi[8], Hiromasa Ohira[1]**

[1]Department of Gastroenterology, Fukushima Medical University, School of Medicine, Fukushima, Japan; [2]Department of Gastroenterology, Fukushima Rosai Hospital, Iwaki, Japan; [3]Department of Gastroenterology, Aizu Medical Center, Fukushima Medical University, Aizu, Japan; [4]Department of Gastroenterology, Ohta Nishinouchi Hospital, Koriyama, Japan; [5]Department of Gastroenterology, Fukushima Redcross Hospital, Fukushima, Japan; [6]Department of Gastroenterology, Soma General Hospital, Soma, Japan; [7]Department of Gastroenterology, Saiseikai Fukushima General Hospital, Fukushima, Japan; [8]Department of Endoscopy, Fukushima Medical University Hospital, Fukushima, Japan

**\*For correspondence:**
kitachuuou335@yahoo.co.jp

**Competing interest:** The authors declare that no competing interests exist.

## eLife Assessment

This **valuable** study discusses a hot topic in post-endoscopic retrograde cholangiopancreatography pancreatitis. The new score for predicting post-ERCP pancreatitis offers an idea about the risk of pancreatitis before the procedure. Although most scores depend on intraprocedural maneuvers, such as the number of attempts to cannulate the papilla, this is a **solid** retrospective single-center study in one country. To be validated in the future, this score will need to be done in many countries and on large numbers of patients.

## Abstract

**Background:** Post-endoscopic retrograde cholangiopancreatography (ERCP) pancreatitis (PEP) is a severe and deadly adverse event following ERCP. The ideal method for predicting PEP risk before ERCP has yet to be identified. We aimed to establish a simple PEP risk score model (SuPER model: Support for PEP Reduction) that can be applied before ERCP.

**Methods:** This multicenter study enrolled 2074 patients who underwent ERCP. Among them, 1037 patients each were randomly assigned to the development and validation cohorts. In the development cohort, the risk score model for predicting PEP was established via logistic regression analysis. In the validation cohort, the performance of the model was assessed.

**Results:** In the development cohort, five PEP risk factors that could be identified before ERCP were extracted and assigned weights according to their respective regression coefficients: –2 points for pancreatic calcification, 1 point for female sex, and 2 points for intraductal papillary muci-nous neoplasm, a native papilla of Vater, or the pancreatic duct procedures (treated as 'planned

pancreatic duct procedures' for calculating the score before ERCP). The PEP occurrence rate was 0% among low-risk patients (≤0 points), 5.5% among moderate-risk patients (1–3 points), and 20.2% among high-risk patients (4–7 points). In the validation cohort, the C statistic of the risk score model was 0.71 (95% CI 0.64–0.78), which was considered acceptable. The PEP risk classification (low, moderate, and high) was a significant predictive factor for PEP that was independent of intraprocedural PEP risk factors (precut sphincterotomy and inadvertent pancreatic duct cannulation) (OR 4.2, 95% CI 2.8–6.3; p<0.01).

**Conclusions:** The PEP risk score allows an estimation of the risk of PEP prior to ERCP, regardless of whether the patient has undergone pancreatic duct procedures. This simple risk model, consisting of only five items, may aid in predicting and explaining the risk of PEP before ERCP and in preventing PEP by allowing selection of the appropriate expert endoscopist and useful PEP prophylaxes.

**Funding:** No external funding was received for this work.

## Introduction

Endoscopic retrograde cholangiopancreatography (ERCP) is widely performed as an important diagnostic and therapeutic procedure for pancreaticobiliary diseases. Among endoscopic procedures, ERCP-related procedures are relatively risky. The high-risk adverse events of ERCP include duodenal perforation and bleeding after endoscopic sphincterotomy (EST) and post-ERCP pancreatitis (PEP). The rate of PEP occurrence is 3.1–13.0% (*Andriulli et al., 2007*; *Freeman et al., 1996*; *Glomsaker et al., 2013*; *Katsinelos et al., 2014*; *Kochar et al., 2015*; *Loperfido et al., 1998*). PEP can even become life-threatening. The fatality rate of PEP is 0.1–0.7% (*Andriulli et al., 2007*; *Kochar et al., 2015*). Therefore, the decision to perform ERCP should be made carefully, considering each patient's risk factors for PEP.

To predict an individual patient's PEP risk, six scoring systems have been devised (*Archibugi et al., 2023*; *Chiba et al., 2021*; *DiMagno et al., 2013*; *Friedland et al., 2002*; *Fujita et al., 2021*; *Zheng et al., 2020*). The first risk scoring system for PEP occurrence was established in 2002. In that study, pain during the procedure, pancreatic duct cannulation, a history of PEP, and the number of cannulation attempts were identified as risk factors for PEP. After the first scoring system was reported, each new scoring system used risk factors that were extracted via multivariate analyses. These included various patient characteristics before ERCP and postprocedural risk factors. Postprocedural risk factors, such as precut sphincterotomy, procedure time, and difficult cannulation, have been proposed, but it is difficult to predict these risk factors and determine the PEP risk before ERCP. Thus, a new prediction scoring system for PEP before ERCP is desirable. If the risk of PEP can be predicted before ERCP, then the expert endoscopist can perform ERCP from the start, and high-PEP-risk procedures (e.g., precut sphincterotomy, multiple cannulation attempts, and inadvertent pancreatic duct cannulation) can be avoided (*Testoni et al., 2010*; *Wang et al., 2009*). If biliary cannulation without the use of at least one high-PEP-risk procedure is difficult, other treatments (e.g., percutaneous transhepatic biliary drainage [PTBD] or endoscopic ultrasound [EUS]-guided biliary drainage [EUS-BD]) could be considered.

Therefore, we aimed to establish a PEP prediction model using only risk factors that can be gathered before ERCP. Our model was developed and validated with multicenter data from Japan.

## Methods

We performed a multicenter retrospective study at six institutions in Japan. This study was approved by the institutional review board of Fukushima Medical University and that of each partner medical institution (number 2453). The analysis used anonymous clinical data obtained after all the participants agreed to treatment with written consent; thus, patients were not required to provide informed consent for the study. The details of the study can be found on the homepage of Fukushima Medical University.

### Patients

Among 2176 patients who underwent ERCP between November 2020 and October 2022, 2074 were included in this study. The other 102 patients were excluded for the following reasons: history of choledochojejunostomy, acute pancreatitis, choledochoduodenal fistula, difficulty finding the Vater

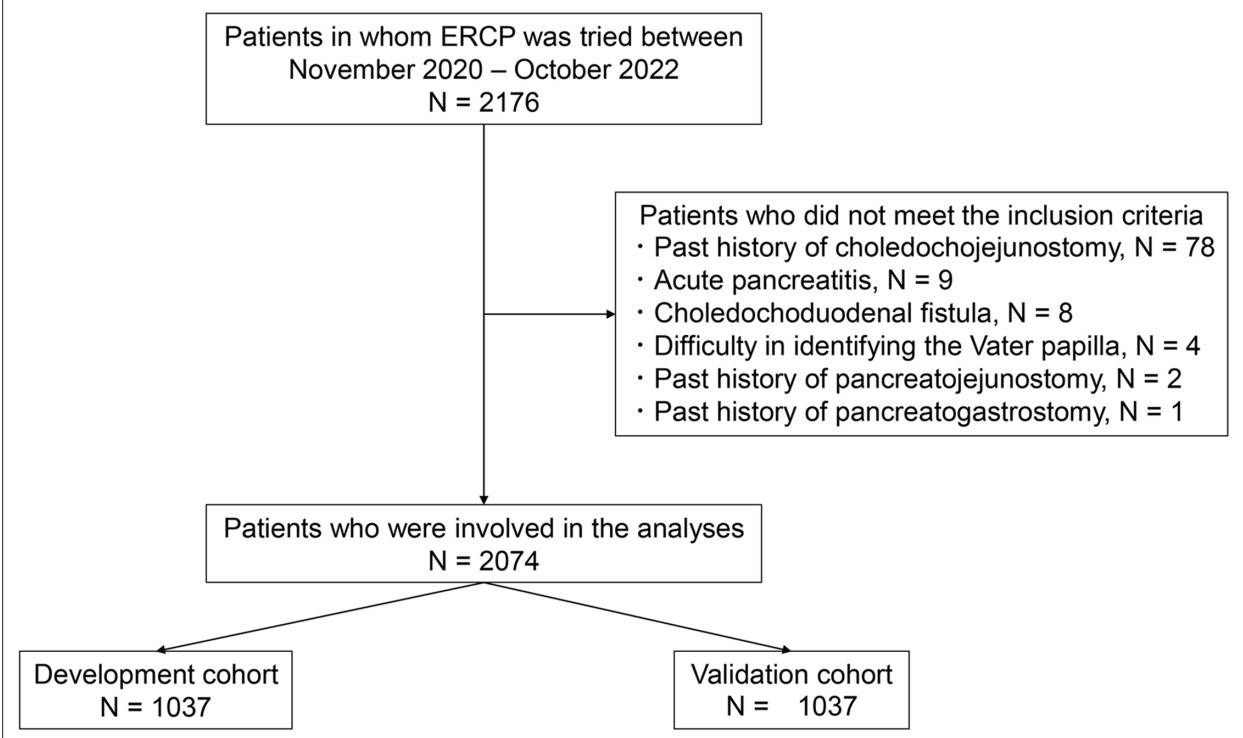

**Figure 1.** Flowchart of the inclusion criteria. ERCP, endoscopic retrograde cholangiopancreatography.

papilla, history of pancreatojejunostomy, or history of pancreatogastrostomy (**Figure 1**). The predictive PEP score is not necessary for the excluded patients. The reasons were as follows. Biliary duct cannulation was not attempted in patients for whom it was difficult to identify the Vater papilla. The biliary tract was separated from the pancreas in patients with a history of choledochojejunostomy, pancreatojejunostomy, or pancreatogastrostomy. PEP risk was thought to be clearly low in these patients and patients who underwent bile duct cannulation via the choledochoduodenal fistula. PEP diagnosis is difficult in patients with acute pancreatitis, whose diagnosis is currently in progress.

## Study design

We randomly sampled 50% of the patients as the development cohort and 50% as the validation cohort (**Figure 1**). In the development cohort, we established a risk scoring system for predicting PEP before ERCP, which was named the support for PEP reduction model (SuPER model). The validation cohort was used to confirm the effectiveness of the scoring system. PEP diagnosis and severity were assessed according to Cotton's criteria (**Cotton et al., 1991**). Patients who experienced abdominal pain and had hyperamylasemia (more than three times the normal upper limit) at least 24 hr after ERCP were diagnosed with PEP. Mild PEP was defined as pancreatitis that required prolongation of the planned hospitalization by 2–3 days. Moderate PEP was defined as pancreatitis that required 4–10 days of hospitalization. Severe PEP was defined as pancreatitis that required more than 10 days of hospitalization or intervention or hemorrhagic pancreatitis, phlegmon, or pseudocysts.

To establish the risk score, the risk factors for PEP were investigated via data from the development cohort. To determine the PEP risk score, factors that might be associated with PEP occurrence were investigated. To predict the PEP risk score before ERCP, factors related to patient characteristics and previously scheduled procedures, as reported in the Japanese guidelines for acute pancreatitis and PEP, were selected (**Itoi et al., 2024**; **Takada et al., 2022**). The patients' risk factors included age <50 years, female sex, a history of pancreatitis, a history of PEP, a history of gastrectomy, pancreatic cancer, intraductal papillary mucinous neoplasm (IPMN), a native papilla of Vater, absence of chronic pancreatitis (CP), normal serum bilirubin (≤1.2 mg/dl), and periampullary diverticulum (**Ding et al., 2015**; **Freeman et al., 2001**; **Freeman et al., 1996**; **Fujita et al., 2022**; **Fujita et al., 2021**; **Masci et al., 2003**; **Wang et al., 2009**; **Williams et al., 2007**; **Zheng et al.,**

*2020*). Pancreatic divisum was excluded from the patient risk factor list because pancreatic divisum was observed in only two patients. Pancreatic calcification and a diameter of the main pancreatic duct >3 mm were considered to indicate CP (*Beyer et al., 2023*; *Sarner and Cotton, 1984*). These imaging findings were confirmed by CT, MRI, and EUS before ERCP. The CT and MRI findings were reviewed by radiologists. IPMNs were diagnosed according to the results of CT, MRI, and EUS. As pre-ERCP prophylaxes for PEP, protease inhibitors (gabexate mesilate or nafamostat mesilate), intravenous hydration, and NSAID suppositories have been used (*Fujita et al., 2022*). As planned procedure-related risk factors, EST, endoscopic papillary balloon dilation (EPBD), endoscopic papillary large balloon dilation (EPLBD) using a ≥12 mm balloon catheter (*Itoi et al., 2018*), biliary stone removal, ampullectomy, biliary stent material (plastic stent, self-expandable metallic stent [SEMS], or covered SEMS [CSEMS]), inside stent placement, and procedures on the pancreatic duct were evaluated (*Freeman et al., 2001*; *Freeman et al., 1996*; *Harewood et al., 2005*; *Kato et al., 2022*; *Masci et al., 2003*; *Masci et al., 2001*; *Testoni et al., 2010*; *Williams et al., 2007*). A biliary stent above the Vater papilla was also assessed as a prophylactic measure against PEP (*Ishiwatari et al., 2013*).

To demonstrate the independence of the established risk classification, the relationships between it and intraprocedural PEP risk factors (including precut sphincterotomy and inadvertent pancreatic duct cannulation) (*Testoni et al., 2010*; *Wang et al., 2009*) were investigated. Because of the retrospective nature of the data, the exact number of cannulations and the number of cannulation attempts were not available. Therefore, multiple cannulation attempts and a prolonged cannulation time could not be investigated as intraprocedural PEP risk factors.

## Sample size

The primary aim of this study was to establish a PEP prediction model that could be used to calculate a risk score before ERCP. To construct a prediction model via logistic regression analysis, 10 events per explanatory variable were needed (*Wynants et al., 2015*). Seven variables were evaluated in the development cohort, so 70 PEP patients were included. Five variables were evaluated in the validation cohort, so 50 PEP patients were necessary. According to a previous systematic review, the rate of PEP occurrence was 9.7% (*Kochar et al., 2015*). Therefore, at least 722 and 521 patients were included in the development and validation cohorts, respectively.

## Statistical analysis

In the development cohort, univariate and multivariate logistic regression analyses were performed to identify the risk factors for PEP. The factors that had a p-value<0.10 in the univariate analysis were included in the multivariate analysis. To construct the scoring system for PEP risk, the factors with p<0.10 in the multivariate analysis were ultimately included in the risk score model. The factors selected in the multivariate analysis were assigned points according to the regression coefficient (each variable's risk points = the ratio of the variable's regression coefficient/minimum regression coefficient). The sum of the assigned points was calculated for each patient, and the patients were classified into three groups (low risk, moderate risk, and high risk) according to the expected rate of PEP occurrence (*Friedland et al., 2002*). The risk classification system (SuPER model) was also applied to the validation cohort.

With respect to both the development and validation cohorts, the effectiveness of the risk score model was evaluated as follows. The correlations between the risk score, risk classification, and PEP occurrence were evaluated via the Cochran–Armitage trend test. The predictive accuracy of the risk score was assessed via the C statistic. The goodness of fit of the model was evaluated via the Hosmer–Lemeshow test. The independence of the established risk classification from the unexpected intraprocedural PEP risk factors was assessed via multivariate logistic regression analyses.

Patients with missing data for variables selected in the risk score model were removed from the final cohort.

Statistical analyses were performed using EZR version 1.62 (Saitama Medical Centre, Jichi Medical University, Saitama, Japan) and SPSS version 26.0 (IBM Corp., Armonk, NY). A p-value<0.05 indicats statistical significance.

## Results

### Patient characteristics and ERCP outcomes in each cohort

The patient characteristics and ERCP outcomes of each cohort are shown in *Table 1*. A total of 1037 patients were assigned to each of the development and validation cohorts, including 70 (6.8%) and 64 (6.2%) patients diagnosed with PEP, respectively. Age, history of pancreatitis, and procedures involving the pancreatic duct were significantly different between the development cohort and the validation cohort. The pre-ERCP prophylactic measures used at each hospital differed, and not all patients received prophylaxis.

### Construction of the PEP risk scoring system

According to the univariate analyses, age <50 years, female sex, IPMN, a native papilla of Vater, pancreatic calcification, EST, and procedures on the pancreatic duct had p-values<0.10 (*Table 2*). According to the multivariate analysis, female sex, IPMN, a native papilla of Vater, pancreatic calcification, and procedures on the pancreatic duct had p-values<0.10. These factors were assigned risk points according to their respective regression coefficients.

The risk score of each patient was calculated as the total number of risk points and ranged from –2 to 7 points (*Table 3*). The risk score was found to be correlated with PEP occurrence (p<0.01, Cochran–Armitage trend test). The patients were classified as low (≤0 points), moderate (1–3 points), or high risk (4–7 points) for PEP according to the risk score. The PEP rates were 0% (0/327) among the low-risk patients, 5.5% (27/492) among the moderate-risk patients, and 20.2% (39/193) among the high-risk patients. The risk classification was correlated with PEP occurrence (p<0.01, Cochran–Armitage trend test).

The C statistic of the risk score model was sufficiently high at 0.77 (95% CI 0.72–0.82) (*Table 4*). The goodness of fit of the risk score model was also confirmed by the Hosmer–Lemeshow test (p=0.59).

### Validation of the PEP risk scoring system

The risk score was associated with PEP occurrence in the validation cohort (p<0.01, Cochran–Armitage trend test) (*Table 3*). We found that 2.4% (8/331) of the patients at low risk, 5.3% (27/513) of those at moderate risk, and 18.0% (29/161) of those at high risk experienced PEP. The risk classification was also correlated with PEP occurrence in the validation cohort (p<0.01, Cochran–Armitage trend test).

The C statistic of the risk score was 0.71, which was also high in the validation cohort (*Table 4*). The PEP risk score model showed good fitness according to the Hosmer–Lemeshow test (p=0.40). According to the above results, the preprocedural PEP risk can be calculated, as shown in *Figure 2*. The pancreatic duct procedure was treated as a 'planned pancreatic duct procedure'. Otherwise, we can calculate both the PEP risk score, which involves the pancreatic duct procedure, and the score, which does not involve the pancreatic duct procedure.

### Risk classification and unexpected PEP risk factors

The relationships between the established risk classification and intraprocedural PEP risk factors are shown in *Supplementary file 1*. For all patients in the development cohort and the validation cohort, the risk classification was significantly associated with the occurrence of PEP. On the other hand, precut sphincterotomy and inadvertent pancreatic duct cannulation were not significantly associated with the occurrence of PEP.

## Discussion

In this multicenter study, we aimed to establish a preprocedural PEP prediction model. As a result, we created a risk scoring system (the SuPER model) using five items that could be measured before performing ERCP. The term 'pancreatic duct procedures' was 'planned pancreatic duct procedures' in the model. When unintended pancreatic duct procedures are considered, we can calculate both the PEP risk score with pancreatic duct procedures and that without pancreatic duct procedures. With this score, PEP occurrence can be accurately predicted to some degree. Although the development cohort and validation cohort had significantly different patient characteristics and ERCP outcomes, the risk score was good in both cohorts. In addition, the established PEP risk classification was associated with PEP occurrence independent of unpredictable intraprocedural PEP risk procedures.

**Table 1.** Comparison of patient characteristics and ERCP outcomes between the development and validation cohorts.

| | Development cohort (n=1037) | Validation cohort (n=1037) | p-Value |
|---|---|---|---|
| *Patient factors* | | | |
| Age, years, mean ± SD | 73.8 ± 12.7 | 75.1 ± 12.5 | 0.02 |
| Sex, n, male/female | 642/395 | 629/408 | 0.59 |
| History of pancreatitis, n (%) | 73 (7.0) | 45 (4.4) | 0.01 |
| History of PEP, n (%) | 26 (2.5) | 24 (2.3) | 0.89 |
| History of gastrectomy, n (%) | 82 (7.9) | 88 (8.5) | 0.69 |
| Billroth-I reconstruction, n | 24 | 25 | |
| Billroth-II reconstruction, n | 23 | 25 | |
| Roux-en-Y reconstruction, n | 33 | 36 | |
| Double tract, n | 1 | 1 | |
| Gastric tube reconstruction, n | 1 | 1 | |
| Pancreatic cancer, n (%) | 145 (14.0) | 174 (16.8) | 0.09 |
| IPMN, n (%) | 17 (1.6) | 8 (0.8) | 0.11 |
| Native papilla of Vater, n (%) | 535 (51.6) | 494 (47.7) | 0.08 |
| Total bilirubin, mg/dl, mean ± SD * | 3.5 ± 5.3 | 3.6 ± 5.0 | 0.45 |
| Diameter of the MPD, mm, mean ± SD † | 2.84 ± 2.63 | 3.1 ± 2.9 | 0.10 |
| Pancreatic calcification, n (%) ‡ | 107 (10.6) | 87 (8.7) | 0.15 |
| Periampullary diverticulum, n (%) | 207 (20.0) | 224 (21.6) | 0.39 |
| *Pre-ERCP prophylaxis* | | | |
| Protease inhibitors, n (%) | 709 (68.4) | 703 (67.8) | 0.81 |
| Intravenous hydration, n (%) | 22 (2.1) | 14 (1.4) | 0.24 |
| NSAID suppository, n (%) | 53 (5.1) | 45 (4.3) | 0.47 |
| *Factors related to the planned procedure* | | | |
| EST, n (%) | 449 (43.3) | 434 (41.9) | 0.53 |
| EPBD, n (%) | 31 (3.0) | 40 (3.9) | 0.33 |
| EPLBD, n (%) | 56 (5.4) | 55 (5.3) | 1.0 |
| Biliary stone removal, n (%) | 327 (31.5) | 342 (33.0) | 0.51 |
| Ampullectomy, n (%) | 5 (0.5) | 5 (0.5) | 1.0 |
| Biliary stent, n (%) | 594 (57.3) | 611 (58/9) | 0.48 |
| Plastic stent, n (%) | 445 (42.9) | 436 (42.0) | 0.72 |
| SEMS, n (%) | 119 (11.5) | 122 (11.8) | 0.89 |
| CSEMS, n (%) | 36 (3.5) | 44 (4.2) | 0.43 |
| Biliary stent above the papilla, n (%) | 45 (4.3) | 47 (4.5) | 0.92 |
| Procedures on the pancreatic duct, n (%) | 285 (27.5) | 237 (22.9) | 0.017 |
| *PEP occurrence,* n (%) | 70 (6.8) | 64 (6.2) | 0.66 |
| Mild, n | 60 | 53 | |
| Moderate, n | 8 | 7 | |
| Severe, n | 2 | 4 | |

ERCP, endoscopic retrograde cholangiopancreatography; PEP, post-ERCP pancreatitis; IPMN, intraductal papillary mucinous neoplasm; MPD, main pancreatic duct; EST, endoscopic sphincterotomy; EPBD, endoscopic papillary balloon dilation; SEMS, self-expandable metallic stent; CSEMS, covered SEMS.

*Data were available for 2042 patients.

†Data were available for 1671 patients.

‡Data were available for 2017 patients.

**Table 2.** Logistic regression analysis of predictive factors for PEP in the development cohort.

| | Univariate analysis | | | Multivariate analysis | | | | |
|---|---|---|---|---|---|---|---|---|
| | OR | 95% CI | p- Value | OR | 95% CI | p-Value | Regression coefficient | Points |
| Age <50 years | 2.42 | 0.99–6.0 | 0.053 | 1.76 | 0.67–4.63 | 0.25 | 0.56 | - |
| Female | 1.91 | 1.17–3.10 | <0.01 | 1.72 | 1.03–2.89 | 0.039 | 0.55 | 1 |
| History of pancreatitis | 1.26 | 0.53–3.0 | 0.61 | | | | | |
| History of PEP | 1.84 | 0.54–6.28 | 0.33 | | | | | |
| History of gastrectomy | 0.89 | 0.35–2.27 | 0.81 | | | | | |
| Pancreatic cancer | 1.03 | 0.51–2.06 | 0.94 | | | | | |
| IPMN | 8.15 | 2.92–22.7 | <0.01 | 3.04 | 0.97–9.52 | 0.056 | 1.11 | 2 |
| Native papilla of Vater | 4.49 | 2.42–8.30 | <0.01 | 2.72 | 1.30–5.71 | <0.01 | 1.0 | 2 |
| Total bilirubin ≤1.2 mg/dl * | 1.13 | 0.69–1.84 | 0.62 | | | | | |
| Diameter of the MPD >3 mm† | 1.31 | 0.76–2.25 | 0.33 | | | | | |
| Pancreatic calcification‡ | 0.36 | 0.11–1.17 | 0.089 | 0.32 | 0.10–1.1 | 0.072 | −1.13 | -2 |
| Periampullary diverticulum | 0.65 | 0.33–1.30 | 0.22 | | | | | |
| Protease inhibitors | 0.72 | 0.44–1.19 | 0.20 | | | | | |
| Intravenous hydration | 1.39 | 0.32–6.08 | 0.66 | | | | | |
| NSAID suppository before ERCP | 1.47 | 0.57–3.83 | 0.43 | | | | | |
| EST | 1.71 | 1.05–2.79 | 0.03 | 0.83 | 0.45–1.52 | 0.54 | −0.19 | - |
| EPBD | <0.01 | 0–infinity | 0.98 | | | | | |
| EPLBD | 0.24 | 0.03–1.76 | 0.16 | | | | | |
| Biliary stone removal | 0.68 | 0.39–1.19 | 0.18 | | | | | |
| Ampullectomy | 3.49 | 0.39–31.6 | 0.27 | | | | | |
| Biliary stent | 0.93 | 0.57–1.52 | 0.78 | | | | | |
| Plastic stent | 0.72 | 0.44–1.20 | 0.21 | | | | | |
| SEMS | 1.66 | 0.87–3.20 | 0.13 | | | | | |
| CSEMS | 0.81 | 0.19–3.43 | 0.77 | | | | | |
| Biliary stent above the papilla | 0.30 | 0.04–2.24 | 0.24 | | | | | |
| Procedures on the pancreatic duct | 4.77 | 2.89–7.89 | <0.01 | 3.49 | 1.99–6.12 | <0.01 | 1.25 | 2 |

PEP, post–endoscopic retrograde cholangiopancreatography pancreatitis; IPMN, intraductal papillary mucinous neoplasm; MPD, main pancreatic duct; EST, endoscopic sphincterotomy; EPBD, endoscopic papillary balloon dilation; EPLBD, endoscopic papillary large balloon dilation; SEMS, self-expandable metallic stent; CSEMS, covered SEMS.

*Data were available for 1024 patients in the development cohort.

†Data were available for 985 patients in the development cohort.

‡Data were available for 1012 patients in the development cohort.

This risk scoring and classification of PEP have several advantages. First, the score is calculated using only five items, all of which can be easily assessed via medical interviews and imaging (e.g., CT). One scoring system included sphincter of Oddi dysfunction (SOD) as a test item (*DiMagno et al., 2013*). The diagnosis of SOD requires sphincter of Oddi manometry and fulfillment of the criteria for biliary pain, but sphincter of Oddi manometry is not widely used (*Cotton et al., 2016*). The diagnostic criterion for biliary pain included 8 items, and that for SOD included 15 items. Among the items of the SuPER risk scoring system, pancreatic calcification was assigned –2 points. Its low weighting could be explained by the following. The international conceptual model of CP can be divided into four stages: acute pancreatitis–recurrent acute pancreatitis, early CP, established CP, and end-stage CP (*Whitcomb et al., 2016*). Established CP patients have already passed the acute pancreatitis–recurrent acute pancreatitis course, and pancreatic calcification has been reported in established CP patients. Acinar dysfunction has also been observed in these patients (*Whitcomb et al., 2016*). Therefore, patients with pancreatic calcification may have a lower incidence of PEP.

Second, the SuPER risk score can be determined before the ERCP procedure, as the established risk classification was found to be the sole significant factor predicting the occurrence of PEP independent

**Table 3.** Patient distribution in terms of risk score and classification.

| Risk score | | Development cohort (n=1012) * | | | Validation cohort (n=1005) † | | |
|---|---|---|---|---|---|---|---|
| | | PEP occurrence, N | PEP rate (95% CI) (%) | p-Value ‡ | PEP occurrence, N | PEP rate (95% CI) (%) | p-Value ‡ |
| -2 | | 0/29 | 0 (0–11.9) | <0.01 | 0/33 | 0 (0–10.6) | <0.01 |
| -1 | | 0/9 | 0 (0–33.6) | | 0/5 | 0 (0–52.2) | |
| 0 | | 0/289 | 0 (0–1.3) | | 8/293 | 2.7 (1.2–5.3) | |
| 1 | | 6/140 | 4.3 (1.6–9.1) | | 5/160 | 3.1 (1.0–7.1) | |
| 2 | | 8/202 | 4.0 (1.7–7.7) | | 14/195 | 7.2 (4.0–11.8) | |
| 3 | | 13/150 | 8.7 (4.7–14.4) | | 8/158 | 5.1 (2.2–9.7) | |
| 4 | | 18/97 | 18.6 (11.4–27.7) | | 14/84 | 16.7 (9.4–26.4) | |
| 5 | | 17/83 | 20.5 (12.4–30.8) | | 14/71 | 19.7 (11.2–30.9) | |
| 6 | | 3/9 | 33.3 (7.5–70.1) | | 0/3 | 0 (0–70.8) | |
| 7 | | 1/4 | 25.0 (0.6–80.6) | | 1/3 | 33.3 (0.8–90.6) | |
| Risk classification | Risk score | PEP occurrence, N | PEP rate (95% CI) (%) | p-Value ‡ | PEP occurrence, N | PEP rate (95% CI) (%) | p-Value ‡ |
| Low | ≤0 | 0/327 | 0 (0–1.1) | <0.01 | 8/331 | 2.4 (1.0–4.7) | <0.01 |
| Moderate | 1–3 | 27/492 | 5.5 (3.6–7.9) | | 27/513 | 5.3 (3.5–7.6) | |
| High | 4–7 | 39/193 | 20.2 (14.8–26.6) | | 29/161 | 18.0 (12.4–24.8) | |

PEP, post–endoscopic retrograde cholangiopancreatography pancreatitis.

*There were missing data for 25 patients.

†Data for 32 patients were missing.

‡The correlations between the risk score or classification and PEP occurrence were evaluated via the Cochran–Armitage test.

of intraprocedural PEP risk factors. As described in the Background section, precut sphincterotomy, multiple cannulation attempts, and a cannulation time greater than 10 min were identified as high-risk factors that cannot be accounted for prior to ERCP (*Testoni et al., 2010*; *Wang et al., 2009*). Although the established PEP risk classification was independent of the included intraprocedural risk factors (precut sphincterotomy and inadvertent pancreatic duct cannulation), detailed data on the number of cannulation attempts and the cannulation time were not available. Therefore, to avoid intraoperative procedures associated with a high risk of PEP occurrence, an expert endoscopist can initially perform ERCP for high-PEP-risk patients. In addition, PEP prophylaxis can be administered beforehand for high-PEP-risk patients. As effective prophylaxes for PEP, rectal non-steroidal anti-inflammatory drug (NSAID) and pancreatic stent placement have been reported (*Akshintala et al., 2021*; *Choi et al., 2023*; *Elmunzer et al., 2008*; *Koshitani et al., 2022*; *Murray et al., 2003*; *Sperna Weiland et al., 2022*; *Sugimoto et al., 2019*; *Yang et al., 2020*). In this report, rectal NSAID use was not identified as a significant factor preventing PEP. One reason for this is that in past reports describing the use of rectal NSAIDs to prevent PEP, patients at high risk for PEP were often treated. In contrast, this study included all patients who underwent ERCP. Another reason might be the difference in dose. 100 mg of rectal diclofenac have been used in most past reports, whereas the efficacy of low-dose rectal diclofenac (25–50 mg) for preventing PEP is under discussion (*Maeda et al., 2021*; *Okuno et al., 2018*; *Otsuka et al., 2012*; *Sakai et al., 2023*; *Takaori et al., 2021*; *Tomoda et al., 2021*). In this study, 12.5–50 mg of diclofenac was used. In Japan, the approved diclofenac dose covered by insurance is 50 mg or less, with the dose typically being lower for elderly patients. Therefore, diclofenac doses ranging from 12.5 to 50 mg were prescribed by the doctors depending on the age and size of the patients. Pancreatic stent placement itself is one of the procedures performed on the pancreatic duct and is a higher-risk procedure for PEP than endoscopic biliary procedures without an approach to the pancreatic duct (*Supplementary file 2*). Moreover, pancreatic stent placement has become a

**Table 4.** Goodness of fit of the risk score model.

| | Development cohort | Validation cohort |
|---|---|---|
| C statistic (95% CI) | 0.77 (0.72–0.82) | 0.71 (0.64–0.78) |
| Hosmer–Lemeshow test, p value | 0.59 | 0.40 |

---

## Preprocedural risk score for Post-ERCP pancreatitis

**1. Please check the corresponding items and calculate the total score.**

| Risk factors | Points |
|---|---|
| ☐ Female sex | 1 |
| ☐ Native papilla of Vater | 2 |
| ☐ Pancreatic calcification on imaging | -2 |
| ☐ IPMN | 2 |
| ☐ Planned pancreatic duct procedures | 2 |

Total ☐ points

**2. Please find the predictive post-ERCP pancreatitis rate according to the total score.**

| Total points | Risk group | Predictive post-ERCP pancreatitis rate |
|---|---|---|
| ≤ 0 | Low | 0-2% |
| 1-3 | Moderate | 5% |
| 4-7 | High | 18-20% |

**※The post-ERCP pancreatitis rate might change due to the actual ERCP procedure performed.**

**Figure 2.** Example of the preprocedural PEP risk checklist. ERCP, endoscopic retrograde cholangiopancreatography; IPMN, intraductal papillary mucinous neoplasm; PEP, post-ERCP pancreatitis.

---

prophylactic treatment for PEP in patients who have undergone pancreatography or wire placement to the pancreatic duct (*Mazaki et al., 2014*; *Sugimoto et al., 2019*). As described above, pancreatic stent placement was performed along with high-risk-PEP procedures (i.e., guidewire placement to the pancreatic duct or pancreatography); therefore, pancreatic stent placement was grouped with the other endoscopic retrograde pancreatography procedures as 'procedures on the pancreatic duct'. In a recent randomized trial involving 1950 patients, the combination of rectal NSAIDs and prophylactic pancreatic stents was more effective for preventing PEP than NSAIDs alone (*Elmunzer et al., 2024*). For high-PEP-risk patients with high scores, the combination of prophylactic methods might be desirable.

This study has several limitations. First, the study was retrospective, and there were missing data. However, the results reported are trustworthy. The percentage of patients who did not meet the inclusion criteria was not greater than 5%, and the percentage of missing data was not greater than 1%. As described in the 'Methods' section, patients with missing data for the variables selected in the risk score model were removed from the final cohort. The reliability of the SuPER risk score model was also statistically confirmed. Second, some factors cannot be assessed before ERCP. Additional procedures can be conducted during ERCP, and unplanned pancreatography is often performed in patients who are scheduled for endoscopic cholangiography or biliary treatment. However, the established PEP risk classification was independent of the included intraprocedural risk factors. A planned procedure for accessing the pancreatic duct is listed in the SuPER risk model. Therefore, we can predict the SuPER risk score and classification of patients regardless of whether they have undergone pancreatic duct procedures. Third, this study was performed in a single country. In the future, prospective validation studies over wider geographical regions are needed.

In conclusion, a simple and useful PEP scoring system (SuPER model) with only five clinical items was developed in this multicenter study. This scoring system may aid in predicting and explaining PEP risk and in selecting appropriate prophylaxes for PEP and endoscopic pancreatobiliary procedures for each patient.

## Acknowledgements

We thank all the staff at the Department of Gastroenterology of Fukushima Medical University, the Department of Endoscopy of Fukushima Medical University Hospital, the Department of Gastroenterology of Fukushima Rosai Hospital, the Department of Gastroenterology of Aizu Medical Center, Fukushima Medical University, the Department of Gastroenterology of Ohtanishinouchi Hospital, Koriyama, the Department of Gastroenterology of Fukushima Redcross Hospital, the Department of Gastroenterology of Soma General Hospital, the Department of Gastroenterology of Saiseikai Fukushima General Hospital, and the Gastroenterology Ward of Fukushima Medical University Hospital. We also thank American Journal Experts for providing English language editing services.

## Additional information

### Funding
No external funding was received for this work.

### Author contributions
Mitsuru Sugimoto, Conceptualization, Data curation, Formal analysis, Validation, Investigation, Methodology, Writing – original draft, Project administration, Writing – review and editing; Tadayuki Takagi, Conceptualization, Supervision, Methodology; Tomohiro Suzuki, Conceptualization, Data curation, Methodology; Hiroshi Shimizu, Yuki Nakajima, Yutaro Takeda, Yuki Noguchi, Reiko Kobayashi, Data curation; Goro Shibukawa, Hidemichi Imamura, Conceptualization, Methodology; Hiroyuki Asama, Naoki Konno, Yuichi Waragai, Data curation, Methodology; Hidenobu Akatsuka, Methodology; Rei Suzuki, Takuto Hikichi, Supervision; Hiromasa Ohira, Conceptualization, Supervision, Writing – review and editing

### Author ORCIDs
Mitsuru Sugimoto ⓘ https://orcid.org/0000-0002-4223-613X

### Ethics
The study protocol was reviewed and approved by the Institutional Review Board of Fukushima Medical University (number 2453). The analysis used anonymous clinical data obtained after all the participants agreed to treatment with written consent; thus, patients were not required to provide informed consent for the study. The details of the study can be found on the homepage of Fukushima Medical University.

Joint Public Review: https://doi.org/10.7554/eLife.101604.3.sa1
Author response https://doi.org/10.7554/eLife.101604.3.sa2

## Additional files

### Supplementary files
Supplementary file 1. Risk classification and unpredictable intraprocedural risk factors for PEP (multivariate logistic regression). PEP, post-endoscopic retrograde cholangiopancreatography pancreatitis. [a] Patients with missing data for variables selected in the risk score model were removed.

Supplementary file 2. Risk of PEP following implantation of pancreatic stents (logistic regression). PEP, post-endoscopic retrograde cholangiopancreatography pancreatitis.

MDAR checklist

Source data 1. The dataset was original raw data without personal information. The data was anonymized and deidentified.

### Data availability

All dataset generated or analyzed during this study are shown in the source data 1. The data was original raw data without personal information. Besides, the data was anonymized and deidentified.

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
