## [Editor Report · eLife Assessment]

This **valuable** study discusses a hot topic in post-endoscopic retrograde cholangiopancreatography pancreatitis. The new score for predicting post-ERCP pancreatitis offers an idea about the risk of pancreatitis before the procedure. Although most scores depend on intraprocedural maneuvers, such as the number of attempts to cannulate the papilla, this is a **solid** retrospective single-center study in one country. To be validated in the future, this score will need to be done in many countries and on large numbers of patients.

---

## [Referee Report · Joint Public Review]

Summary:

This work provides a new general tool for predicting post-ERCP pancreatitis before the procedure depending on pancreatic calcification, female sex, intraductal papillary mucinous neoplasm, a native papilla of Vater, or the use of pancreatic duct procedures. Even though it is difficult for the endoscopist to predict before the procedure which case might have post-ERCP pancreatitis, this new model score can help with the maneuver and when the patient is at high risk of pancreatitis, sometimes can be deadly, so experienced endoscopists can do the procedure from the start. This paper provides a model for stratifying patients before the ERCP procedure into low, moderate, and high risk for pancreatitis. To be validated, this score should be done in many countries and on large numbers of patients. Risk factors can also be identified and added to the score to increase rank.

Strengths:

(1) One of the severe complications of endoscopic retrograde cholangiopancreatography procedure is pancreatitis, so investigators try all the time to find a score that can predict which patients will probably have pancreatitis after the procedure. Most scores depend on the intraprocedural maneuver. Some studies discuss the preprocedural score that can predict pancreatitis before the procure. This study discusses a new preprocedural score for post-ERCP pancreatitis.

(2) Depending on this score that identifies low, moderate, and high-risk patients for post-pancreatitis, so from the start, experienced and well-trained endoscopists can do the procedure or can refer patients to tertiary hospitals or use interventional radiology or endoscopic retrograde cholangiopancreatography.

(3) The number of patients in this study is sufficient to analyze data correctly.

Weaknesses:

(1) It is a single-country, retrospective study.

(2) Many cases were excluded, so the score cannot be applied to those patients.

Comments on revised version:

Depending on old references cannot help us know the current situation. What if there are better more recent predictive tools? It would be better to test the validity of that score against, if present, a proven score to check its validity.

---

## [Author Response]

The following is the authors’ response to the original reviews.

**Joint Public review:**
Summary:This work provides a new general tool for predicting post-ERCP pancreatitis before the procedure depending on pancreatic calcification, female sex, intraductal papillary mucinous neoplasm, a native papilla of Vater, or the use of pancreatic duct procedures. Even though it is difficult for the endoscopist to predict before the procedure which case might have post-ERCP pancreatitis, this new model score can help with the maneuver and when the patient is at high risk of pancreatitis, sometimes can be deadly, so experienced endoscopists can do the procedure from the start. This paper provides a model for stratifying patients before the ERCP procedure into low, moderate, and high risk for pancreatitis. To be validated, this score should be done in many countries and on large numbers of patients. Risk factors can also be identified and added to the score to increase rank.

Thank you for reviewing our manuscript. We hope that this score will be validated in other countries from now on.

Strengths(1) One of the severe complications of endoscopic retrograde cholangiopancreatography procedure is pancreatitis, so investigators try all the time to find a score that can predict which patients will probably have pancreatitis after the procedure. Most scores depend on the intraprocedural maneuver. Some studies discuss the preprocedural score that can predict pancreatitis before the procure. This study discusses a new preprocedural score for post-ERCP pancreatitis.

Thank you for evaluating our manuscript and raising a strength of this manuscript.

(2) Depending on this score that identifies low, moderate, and high-risk patients for post-pancreatitis, so from the start, experienced and well-trained endoscopists can do the procedure or can refer patients to tertiary hospitals or use interventional radiology or endoscopic retrograde cholangiopancreatography.

Thank you for evaluating our manuscript and raising a strength of this manuscript.

(3) The number of patients in this study is sufficient to analyze data correctly.

Thank you for evaluating our manuscript and raising a strength of this manuscript.

Weaknesses:(1) It is a single-country, retrospective study.

Thank you for this comment. It’s exactly as you said. This is a limitation (Lines 326-327).

(2) Many cases were excluded, so the score cannot be applied to those patients.

Thank you for this valuable comment. The predictive PEP score is not necessary for the excluded patients. The reasons were as follows. Biliary duct cannulation was not attempted in patients for whom it was difficult to identify the Vater papilla. The biliary tract was separated from the pancreas in patients with a past history of choledochojejunostomy, pancreatojejunostomy, or pancreatogastrostomy. PEP risk was thought to be low in these patients and patients who underwent bile duct cannulation via the choledochoduodenal fistula. PEP diagnosis is difficult in patients with acute pancreatitis, whose diagnosis is currently in progress. We added these explanations (Lines 98-106).

(3) Many other studies, e.g., https://link.springer.com/article/10.1007/s00464-021-08491-1, https://pubmed.ncbi.nlm.nih.gov/36344369/, that have been published before discussing the same issue, so what is the new with this score?

Thank you for raising the new reference written by Archibugi et al. in 2023. The novelty of our score is that it is calculated using the factors that are investigated before ERCP procedures. The study written by Archibugi et al. involved procedure time and cannulation attempts for PEP prediction. These two factors are unknown before ERCP procedures. Therefore, a preprocedural predictive risk model for PEP was not created before our study was performed. We added the content of the past study written by Archibugi and included the report as a reference (Lines 65-67, 73-74).

(4) The discussion section needs reformulation to express the study's aim and results.

Thank you for this valuable comment. I have rewritten the first paragraph of the discussion. In the paragraph, we showed that the study achieved the aim on the basis of the results (Lines 245-255).

(5) Why did the authors select these items in their scoring system and did not add more variables?

Thank you for this valuable comment. We selected the items listed in the Japanese guidelines for acute pancreatitis and post-ERCP pancreatitis. We added this description (Lines 123-126). The original references of the guidelines were cited in the first draft version.

**Recommendations for the authors:**

**Reviewer #1 (Recommendations for the authors):**
Comment1. Please revise these documents: copyright, disclaimer, ethics approval, consent to participate, consent for publication, data and material availability, competing interests, funding, authors' contributions, and acknowledgments.

First, thank you for reviewing our manuscript. We have already described the required information in the “author information” section. The sentences containing this information were proofread in English.

**Reviewer #2 (Recommendations for the authors):**
Comment 1. It would be best if you did this study in a Prospective way for more validation.

First, thank you for reviewing our manuscript. We have revised our manuscript according to your comments. It’s exactly as you said. These points are limitations (Lines 312-318, lines 326-327). We hope that future validation studies over wider geographic regions will prove our opinions.

Comment 2. The model name should be Acronyum (the first letter of the five items in the risk model).

Thank you for this valuable comment. Sorry, we could not create a memorable model name using the first letter of the five items.

Comment 3. You say that you include the pre-procedure criteria that predict PEP. You state one of the items, pancreatic duct procedure. Do you mean it is a history?

Thank you for this valuable comment. This means that the main purpose is the pancreatic duct. Therefore, the pancreatic duct procedure is listed as “planned pancreatic duct procedures” in Figure 2 (Lines 40-41, 231-234). When an unintended pancreatic duct procedure is performed, we can calculate the risk score by adding two points for “planned pancreatic duct procedures” (Lines 48-49, 247-250).

Comment 4. Regarding calcification, do you mean chronic pancreatitis? It needs more clarification regarding its degree.

Thank you for this valuable comment. We regard pancreatic calcification as a finding of chronic pancreatitis. Pancreatic calcification was defined as the degree that was confirmed by imaging, such as CT, MRI, and EUS. These definitions have been written in the first draft version (Lines 134-137).

Comment 5. Why don't you include young age in the model? Your result found that age less than 50 is significantly associated with PEP.

Thank you for this valuable comment. We selected the PEP risk factors listed in the Japanese guidelines for acute pancreatitis and post-ERCP pancreatitis. Age less than 50 years was listed as a PEP risk factor in the Japanese guidelines for acute pancreatitis. We added this description (Lines 123-126).

Comment 6. There is an ancient reference, some of them in 1994,1996.

Sorry for the old references. These references were written by Cotton et al. 1991, Freeman et al. 1996, and Loperfido et al. 1998. These are still important today. The diagnostic criteria for PEP were determined in the report written by Cotton et al., which is Cotton’s criteria. The other two references are representative reports that described risk factors for PEP, and these two reports were cited in the Japanese guidelines for pancreatitis written by Takada et al. 2022 (Lines 123-126).

Comment 7. In the introduction, you say that the first score includes one of the items for PEP pain during the procedure. It is a little bit strange.

Thank you for this comment. The first PEP risk score did not involve PEP pain but involved pain during the procedure (Line 68).

Comment 8. We know that once ERCP is indicated, you justify the importance of the risk model, stating that if one or more risks are found, we can do EUS or PTD. It is not reasonable to abort the procedure in case of frequent pancreatic duct cannulation or cancel ERCP if pt has one or more risk factors.

Thank you for this valuable comment. If ERCP is performed for high-risk patients, prophylaxes for PEP, such as procedures by experts, pancreatic stent placement, and NSAID suppository insertion, should be performed as much as possible (Lines 281-287, 308-311).

Comment 9. Regarding ERCP pancreatitis criteria, does it include amylase 3t or lipase?

Thank you for this comment. We used Cotton’s criteria for diagnosing PEP. Cotton’s criteria include hyperamylasemia (more than three times the normal upper limit) at least 24 hours after ERCP (114-116).

Comment 10. It is well known that pr with functional biliary disorder has a high incidence of PEP; it doesn't need a manometer for diagnosis. It needs to be included.

Thank you for this comment. Moreover, functional biliary disorders are difficult to diagnose before ERCP procedures (Lines 259-262). The factor that is not apparent before ERCP could not be included in the predictive PEP scoring system.

Comment 11: What is gabexare and nafamost.

Thank you for this comment, and sorry for our insufficient explanation. These compounds include gabexate masilate and nafamostat masilate, which are protease inhibitors. In some institutions, protease inhibitors are used as prophylaxis for PEP. We added “protease inhibitors” (Lines 138-139, Tables 1 and 2).

**Reviewer #3 (Recommendations for the authors):**
Comment 1. The sample size needs clarification.

First, thank you for reviewing our manuscript. The sample size has been included in the “Methods” section (Lines 157-165).

Comment 2. They need to be mentioned cause they depend on old references in discussion and background.

Thank you for this comment. The previous references were written by Cotton et al. 1991, Freeman et al. 1996, and Loperfido et al. 1998. These are still important today. The diagnostic criteria for PEP were determined in the report written by Cotton et al., which is Cotton’s criteria. The other two references are representative reports that described risk factors for PEP, and these two reports were cited in the Japanese guidelines for pancreatitis written by Takada et al. 2022 (Lines 122-126). In the background and discussion, we added new recent references and information related to the references (Lines 65-67, 285-287, 291-295, 308-311).

Comment 3. Case definition should be added to the methodology.

Thank you for this comment. We added patient information. Please refer to the response against the eLife assessment, weakness, (2).

Comment 4. Do you include all who met the inclusion criteria, or was there any random sampling technique?

No, we did not use random sampling techniques.

Comment 5. What is the value of comparing the development and validation groups? I do not think it adds anything new as if you want to exclude confounders. Has the comparison revealed that a confounder does exist? What was your point of view concerning that?

Thank you for this valuable comment, and sorry for the insufficient explanation. The differences between the development cohort and the validation cohort are important because the goodness of fit for the score could be confirmed in significantly different groups. We added this explanation (Lines 197-199, 251-253).